# Monitoring of Sotrovimab-Levels as Pre-Exposure Prophylaxis in Kidney Transplant Recipients Not Responding to SARS-CoV-2 Vaccines

**DOI:** 10.3390/v15081624

**Published:** 2023-07-26

**Authors:** Constantin Aschauer, Andreas Heinzel, Karin Stiasny, Christian Borsodi, Karin Hu, Jolanta Koholka, Wolfgang Winnicki, Alexander Kainz, Helmuth Haslacher, Rainer Oberbauer, Roman Reindl-Schwaighofer, Lukas Weseslindtner

**Affiliations:** 1Division of Nephrology and Dialysis, Department of Medicine III, Medical University of Vienna, 1090 Vienna, Austria; constantin.aschauer@meduniwien.ac.at (C.A.); karin.hu@meduniwien.ac.at (K.H.); jolanta.koholka@akhwien.at (J.K.); wolfgang.winnicki@meduniwien.ac.at (W.W.); alexander.kainz@meduniwien.ac.at (A.K.); rainer.oberbauer@meduniwien.ac.at (R.O.); 2Center of Virology, Medical University of Vienna, 1090 Vienna, Austria; karin.stiasny@meduniwien.ac.at (K.S.); christian.borsodi@meduniwien.ac.at (C.B.); lukas.weseslindtner@meduniwien.ac.at (L.W.); 3Department of Laboratory Medicine, Medical University of Vienna, 1090 Vienna, Austria

**Keywords:** preexposition prophylaxis, monoclonal antibodies, SARS-CoV-2, pharmakokinetics

## Abstract

***Background*** Sotrovimab, a monoclonal antibody against SARS-CoV-2, is used as a pre-exposition prophylaxis (PrEP) against COVID-19, but monitoring strategies using routine test systems have not been defined. ***Methods*** Twenty kidney transplant recipients without antibodies after vaccination received 500 mg Sotrovimab. Antibody levels were quantified over eight weeks using live-virus neutralization (BA1 and BA2), antibody binding assays (TrimericS, Elecsys, QuantiVAC) and surrogate virus neutralization tests (sVNTs; TECOmedical, cPass and NeutraLISA). ***Results*** Sotrovimab neutralized both Omicron subvariants (BA1 NT titer 90 (+−50) > BA2 NT titer 33 (+−15) one hour post infusion). Sotrovimab was measurable on all used immunoassays, although a prior 1:100 dilution was necessary for Elecsys due to a presumed prozone effect. The best correlation with live-virus neutralization titers was found for QuantiVAC and TrimericS, with a respective R^2^ of 0.65/0.59 and 0.76/0.57 against BA1/BA2. Elecsys showed an R^2^ of 0.56/0.54 for BA1/BA2, respectively. sVNT values increased after infusion but had only a poor correlation with live-virus neutralization titers (TECOmedical and cPass) or did not reach positivity thresholds (NeutraLISA). ***Conclusion*** Antibody measurements by the used immunoassays showed differences in antibody levels and only a limited correlation with neutralization capacity. We do not recommend sVNTs for monitoring SARS-CoV-2 neutralization by Sotrovimab.

## 1. Introduction

During the pandemic of severe acute respiratory syndrome coronavirus type 2 (SARS-CoV-2), developing efficient vaccines was the game-changer in mediating protection against severe disease [1]. Nevertheless, not all vaccinated individuals sufficiently responded to the vaccination and produced detectable levels of neutralizing antibodies, even after multiple rounds of vaccination [2]. Such individuals were primarily patients with chronic diseases or on immunosuppressive therapy [3]. 

Monoclonal antibodies (mAbs) against SARS-CoV-2 were developed as a treatment option for severe SARS-CoV-2 infections and prophylaxis following exposure [4,5]. Selected mAbs were further used as a pre-exposition prophylaxis (PrEP) in an off-label application in high-risk indivudials without sufficient immune response following vaccination [4,6,7]. However, Cilgavimab/tixagevimab (Evusheld, AstraZeneca) remains the only mAb combination that was approved for PrEP. 

The majority of these mAbs were less able to neutralize Omicron variants, which displayed significant antigenic changes that inhibited antibody-binding. Thus, a significant reduction in the neutralization capacity against the Omicron BA1 variant (B1.1.529) was observed for most mAbs, including cilgavimab/tixagevimab [8,9,10]. 

Reductions in antibody-mediated neutralization also resulted in a high rate of breakthrough infections with BA1, including the re-occurrence of clinically severe disease cases following cilgavimab/tixagevimab administration [3,4]. However, Sotrovimab (Xevudy, VIR Biotechnology GlaxoSmithKline (Brentford, UK)) retained substantial neutralizing capacities against BA1 and, together with a long half-life of 48.8 days, this made it a suitable PreP candidate in high-risk individuals following the emergence of Omicron BA.1 [7]. 

However, soon the question arose as to which antibody assays were best-suited to quantifying the immediate and long-term efficacy of Sotrovimab infusion. In a previous study, we applied an in-house live-virus neutralization test (NT) and showed that the administration of Sotrovimab resulted in detectable levels of neutralizing antibodies against the Omicron BA1 variant for at least eight weeks [8]. 

Although live-virus NTs are considered the gold standard for measuring antibodies against SARS-CoV-2, these assays are laborious and require high-biosafety-level laboratories [11,12,13,14]. To evaluate other diagnostic tools to quantify antibody levels after Sotrovimab infusion in a PrEP setting, which could also guide re-dosing, we analyzed the antibody kinetics in 20 kidney transplant reciepients (KTRs) without a sufficient response to previous COVID-19 vaccination for eight weeks after infusion of 500 mg Sotrovimab using Omicron-specific BA.1 and BA.2 live-virus NTs and three commercial surrogate NTs (sVNTs; TECOmedical cPass, NeutraLISA) and three commercial Anti-Spike immunoassays [6].

## 2. Materials and Methods

### 2.1. Patients

Twenty KTRs without a sufficient antibody response after multiple vaccinations and no prior infection with SARS-CoV-2 received 500 mg Sotrovimab in a pre-exposure prophylaxis setting. Baseline SARS-CoV-2 antibody levels were all below 15 Binding Antibody Units per milliliter (BAU/mL). Detailed patient characteristics are shown in Table 1. Blood sampling took place before and one hour after infusion, as well as two, four, and eight weeks after infusion. All samples were tested using live-virus NTs, three commercial sVNTs, and three commercial Anti-Spike antibody assays.

### 2.2. Live-Virus Neutralization Test

The live-virus NTs were performed as described previously [15]. In brief, serum samples were incubated at 37 °C with 50–100 tissue culture infectious doses of either Omicron BA1 or BA2 virus strains for one hour. Then, this mixture was applied to a monolayer of VeroE6 cells (ECACC 85020206). After 5 days, the NT titers were assessed by microscope as the reciprocal dilution factor at which serum antibodies prevented a virus cytopathic effect. Serial dilutions ranged from 1:10 to 1:10,240. NT titers ≥ 10 were considered positive. 

### 2.3. Surrogate Virus Neutralization Tests (sVNTs)

All samples were further tested using the following three commercial sVNTs: the cPass (GenScript Biotech, Piscataway Township, NJ, USA), the SARS-CoV-2-NeutraLISA (Euroimmun, Lübeck, Germany), and the TECO SARS-CoV-2 Neutralization Antibody Assay (TECOmedical, Sissach, Switzerland). All these tests were conducted following the manufacturers’ instructions and cut-off values, strictly adhering to the respective protocol and only performing the dilution steps recommended by the manufacturers [11]. sVNTs quantify the antibody-mediated binding inhibition (in %) between angiotensin-converting enzyme 2 (ACE2), the host receptor of SARS-CoV-2, and the viral Receptor Binding Domain (RBD) as a surrogate for the antibodies’ neutralizing abilities [11].

### 2.4. Anti-Spike Immunoassays

Measurements for spike-specific antibodies quantified as BAU/mL were performed using the Anti-SARS-CoV-2 QuantiVAC ELISA (Euroimmun, Lübeck, Germany), the LIAISON^®^ SARS-CoV-2 TrimericS IgG CLIA (DiaSorin, Saluggia, Italy), and the Elecsys Anti-SARS-CoV-2 S (Roche Diagnostics GmbH, Mannheim, Germany). The tests were used following the manufacturers’ recommendations, along with the initially recommended protocols, dilution steps, and cut-off values the manufacturers [11]. Additional dilution steps (1:20, 1:40, 1:60, 1:80, and 1:100) were performed for the LIAISON^®^ SARS-CoV-2 TrimericS IgG CLIA and a single dilution (1:100) for the Elecsys Anti-SARS-CoV-2 S. In principle, these assays measure Spike-specific IgG (QuantiVAC, LIAISON^®^ SARS-CoV-2 TrimericS) or RBD-specific antibodies of all immunoglobulin classes (Elecsys Anti-SARS-CoV-2 S) after binding to the respective target antigen in the assay, either by an enzymatic colorimetric reaction (QuantiVAC) or through (electro)-chemoluminescence (LIAISON^®^ SARS-CoV-2 TrimericS, Elecsys Anti-SARS-CoV-2 S).

### 2.5. Statistical Analysis

The mean, standard deviation and confidence interval were calculated for all antibody levels, and sensitivity and specificity were calculated for the antibody cut-off levels. Ordinary least square (OLS) regression analysis was used for correlation. Due to the lognormal distribution of BA1 and BA2 test results, these data were transformed for regression analysis by logarithm (base 10) before use in regression analysis. 

## 3. Results

### 3.1. Omicron BA1 and BA2 Neutralization Titers for Serum Samples after Sotrovimab Infusion

Before the infusion of Sotrovimab, all KTRs tested negative in the Omicron-BA1- and BA2-specific live-virus NTs (all titers < 10). As shown in Figure 1, one hour after infusion, mean neutralization titers against BA1 peaked at 88 (SD: +−51) and decreased to 40 (SD: +−26), 33 (SD: +−10.60) and 25 (SD: +−16) at two, four and eight weeks, respectively. Overall, live-virus NT titers against BA2 were lower compared to BA1 peaking at 33 (SD: +−15), one hour after infusion. Titers subsequently decreased to 20 (SD: +−9) at week two, and were already negative at week four in all but two patients.

### 3.2. Kinetics of Anti-Spike Antibodies Quantified in BAU/mL

Although the Anti-SARS-CoV-2 QuantiVAC ELISA and the LIAISON^®^ SARS-CoV-2 TrimericS IgG CLIA measured Spike-specific IgG antibodies, both antibody assays showed significant variation in the measured antibody levels in BAU/mL over time after infusion with Sotrovimab.

Antibody levels measured with the QuantiVAC ELISA peaked at 5244 BAU/mL (SD: +−2292) one hour after infusion, and subsequently decreased to 1808.34 BAU/mL (SD: +−526.12) after two weeks, to 1537 BAU/mL (SD: +−459) after four, and to 1048.45 BAU/mL (SD: +−373.30) after eight weeks, respectively (Figure 2A).

Spike-specific IgG antibody levels assessed with the LIAISON^®^ SARS-CoV-2 TrimericS IgG CLIA showed similar kinetics to the antibody levels quantified with the QuantiVAC ELISA, but reached a maximum of 1680.92 BAU/mL (SD: +−643.41) one hour after infusion and decreased to 762 BAU/mL (SD: +−168), 634 BAU/mL (SD: 204) and 536 (SD: +−220) two, four and eight weeks after application.

Notably, the Elecsys Anti-SARS-CoV-2 S ECLIA were measured to have significantly lower antibody levels than the other assays. One hour after Sotrovimab application, the Elecsys Anti-SARS-CoV-2 S ECLIA quantified a mean of 77 BAU/mL (SD: +−39), slightly increasing over time and reaching levels of 530 BAU/mL (SD: +−849) two weeks, 999 BAU/mL (SD: +−597) four weeks, and 1572 BAU/mL (SD: +−1089) eight weeks post-infusion. 

The antibody measurements with the Elecsys Anti-SARS-CoV-2-2 S ECLIA showed high variability between individuals, ranging from >2500 to 145 BAU/mL at two weeks and >2500 BAU/mL to 221 BAU/mL at eight weeks after sotrovimab infusion. Since these measurements were not plausible, we speculated that a prozone effect occurred. Therefore, we performed additional dilution series of the patient serum samples. After an additional 1:100 dilution, the antibody kinetics measured with Elecsys Anti-SARS-CoV-2-2 S ECLIA were similar to the other two assays, with peak values of 18,750 BAU/mL one hour after infusion (SD: +−11,078) followed by a decrease to 9585 BAU/mL (SD: +−3358), 7197 U/mL (SD: +−3049) and 4120 U/mL (SD: +−1767) two, four, and eight weeks after infusion, respectively (Figure 2B).

### 3.3. Antibody Kinetics as Measured by Surrogate Virus Neutralization Tests

Next, we tested the KTRs samples after Sotrovimab PrEP infusion with multiple sVNTs (ACE2-RBD-binding-inhibition assays). As shown in Figure 3, the cPass and the TECO SARS-CoV-2 Neutralization Antibody Assay showed similar kinetics with an increase in ACE2-RBD-binding-inhibition (in %). However, the individual levels of % binding inhibition significantly varied among the two assays. From baseline (pre-infusion) to one-hour post Sotrovimab infusion, the increase was from a mean of 7.43% (SD: +−6.65) to 57.27% (SD: 14.86) for the TECO SARS-CoV-2 Neutralization Antibody Assay and from −13.52% (SD: 10.58) to 42.50% (SD: 11.31) for the cPass, respectively (Figure 3). In consecutive measurements at two, four and eight weeks after Sotrovimab infusion, the levels of ACE2-RBD-inhibition measured by cPass were lower than the ones measured with the TECO SARS-CoV-2 Neutralization Antibody Assay with a median level of 55.80% (SD: 12.15) and 41.08% (SD: 9.40) at week two, 57.56% (SD: 13.49) compared to 41.55% (SD: 9.90) at week four and 56.99% (SD: 15.41) and 41.71% (SD: 11.50) at week eight for the TECO SARS-CoV-2 Neutralization Antibody Assay and cPass, respectively.

Notably, the SARS-CoV-2-NeutraLISA measured significantly lower levels of ACE2-RBD-binding inhibition in the same samples, with median levels of 27.88% (SD: 8.40) one hour after infusion, 25.23% (SD: 8.21) at two weeks, 18.97% (SD: 7.50) at four weeks, and 16.15% (SD: 6.80) at eight weeks after infusion. When using the cut-off levels provided by the manufacturer, these results would be interpreted as negative for neutralizing antibodies (cut-off level of 35%).

### 3.4. Correlation of Titers of Neutralizing Antibodies with Levels of Commercial Assays

Subsequently, the titers of neutralizing antibodies determined in the live-virus NTs of the samples were correlated with the antibody levels measured by the commercial antibody assays. The LIAISON^®^ SARS-CoV-2 TrimericS IgG CLIA and QuantiVAC ELISA Q had the best correlation with NT titers against the BA1 variant and, to a lesser extent, against the BA2 variant (R^2^: 0.65 and 0.59 for QuantiVAC and 0.76 and 0.57 for the TrimericS IgG assay for BA1 and BA2, respectively; Figure 4). Following additional dilution, the correlation between NT titers and the BAU/mL levels measured from the Elecsys Anti-SARS-CoV-2 S ECLIA was weaker, with an R^2^ of 0.56 and 0.54 for BA1 and BA2, respectively.

Importantly, the correlation between live-virus NT titers and levels of ACE2-RBD-binding inhibition after Sotrovimab infusion was very weak for cPass and TECO SARS-CoV-2 Neutralization Antibody Assay, with an R^2^ of 0.46 and 0.10 and 0.46 and 0.11 for BA1 and BA2 for each test respectively. Of note, the NeutraLISA sVNT showed a higher R^2^ of 0.58 and 0.27 for BA1 and BA2, respectively, but the results were all negative, as defined by the manufacturer (Figure 5).

### 3.5. Cut-Off Levels of Antibody Measurements and Live-Virus Neutralization

A 100% specificity for in vitro neutralizing capacity against BA1 in patient sera (defined as NT titer ≥ 10) was observed at antibody levels of 750 BAU/mL on the LIAISON^®^ SARS-CoV-2 TrimericS IgG CLIA, 2100 BAU/mL on the QuantiVAC ELISA and 4060 BAU/mL on the Elecsys assay (after 1:100 dilution). At these values, the sensitivity was 58.1% for the LIAISON^®^ SARS-CoV-2 TrimericS IgG CLIA assay, 43.9% for the QuantiVAC ELISA system and 96.2% for the Elecsys assay (Figure 6).

For the BA2 variant antibody level, cut-offs with 100% specificity for a positive in-vitro neutralization result were 965 BAU/mL for the LIAISON^®^ SARS-CoV-2 TrimericS IgG CLIA, 2538 BAU/mL for the QuantiVAC ELISA and 13,500 BAU/mL for the Elecsys assay (after 1:100 dilution). The corresponding sensitivity values were 48%, 40.8% and 36% for the LIAISON^®^ SARS-CoV-2 TrimericS IgG CLIA assay, the QuantiVAC ELISA system and the Elecsys assay.

We were not able to define a cut-off value for the surrogate NTs as inhibition percentage values persisted at high levels over the entire observational period after Sotrovimab infusion or did not reach the defined thresholds for positivity (Figure 3).

## 4. Discussion

In this study, we analyzed the ability of several readily available high-throughput SARS-CoV-2 antibody measurement systems (three platforms to quantify anti-spike protein antibody levels, i.e., QuantiVAC, TrimericS IgG and Elecsys Anti-SARS-CoV-2 S assay; and three surrogate neutralization tests, i.e., TECO SARS-CoV-2 Neutralization Antibody Assay, cPass and NeutraLISA) to determine Sotrovimab antibody levels in serum samples from patients following PrEP with Sotrovimab. We were able to define anti-spike protein antibody cut-off levels associated with 100% specificity with the in vitro virus neutralization of SARS-CoV-2 Omicron sub-variants that could be used to guide the re-dosing of Sotrovimab. However, some of the evaluated systems showed relevant limitations restricting their use in quantifying antibody levels following Sotrovimab infusion in clinical routine diagnostics. Specifically, the Elecsys system requires an additional dilution (1:100) to avoid inadequately low antibody level read-outs due to the presumed prozone effect caused by the high levels of the monoclonal antibody. Furthermore, all surrogate NT-tests showed poor performance and a low correlation with live-virus NT titers and should not be used to guide Sotrovimab dosing in PrEP. Overall, the measured antibody levels showed a high variability across the different commercial antibody quantification assays, probably due to significant differences in the binding of the monoclonal antibodies to the different target antigens incorporated in the test systems. 

The commercial test systems were compared to live-virus Omicron BA1 and BA2 NTs that represented the dominant SARS-CoV-2 variants circulating in Austria at the beginning of 2022 (Figure 1). The measured NT titers were in line with the literature, pointing to the reduced effectiveness of Sotrovimab against these subvariants, which showed significant antigenic changes affecting antibody binding compared to the ancestral wild-type [8,9,10]. 

Since NTs require high biosafety level laboratories and are laborious, we and others aimed to correlate the antibody levels measured with commercial antibody assays to these NT titers [16]. We observed a high level of variability across all three commercial anti-spike antibody assays when quantifying Sotrovimab in patient serum, despite the normalization of antibody concentrations to the WHO-recommended measuring unit BAU/mL.

Multiple studies have demonstrated such variations [11,17,18]. Indeed, when the same samples from individuals after SARS-CoV-2 infections or vaccinations are comparatively tested, the LIAISON^®^ SARS-CoV-2 TrimericS IgG CLIA usually measures higher BAU/mL levels than the QuantiVAC [11,17]. Interestingly, data from this recent study demonstrated a converse pattern in patients after Sotrovimab infusion, probably due to differences in the binding of the mAbs to the epitopes of the assays´ respective target antigens, whose conformation may vary. Additionally, we found a prozone effect for the Elecsys Anti-SARS-CoV-2 S ECLIA, probably caused by the high level of mAb competing for antigen binding at the same site (Figure 2A). Following an additional dilution step, the antibody levels measured by the Elecsys assay showed the expected kinetics over time, correlated with the antibody profiles assessed by the other assays, confirming the hypothesis [19]. In particular, the prozone effect was most pronounced in the samples obtained one hour after infusion and was diminished at later timepoints. In a previous study, a similar dilution was performed before the Elecsys assay measurements in highly concentrated samples [6].

Notably, we could further show that Sotrovimab strongly affected the results from sVNTs. First, we observed high inter-assay variation that exceeded the variation levels observed in serum samples from convalescent individuals [11]. Furthermore, in contrast to samples from convalescents, we observed higher levels in the TECO SARS-CoV-2 Neutralization Antibody Assay than the cPass [11]. Indeed, while the cPass and TECO SARS-CoV-2 Neutralization Antibody assay provided positive results in all samples, the NeutraLISA failed to detect Sotrovimab neutralization and always tested negative.

Such substantial differences in the results of the sVNTs after Sotrovimab infusion can be explained by the test principle of these assays, which measure the binding inhibition between the viral receptor-binding domain (RBD) and the angiotensin-converting enzyme 2 (ACE2). Since the target binding site of Sotrovimab is a highly conserved epitope outside the ACE2 binding site, this may have significantly affected the readout of these assays [14,20]. Indeed, since sVNTs assess the antibody-mediated RBD-ACE2-binding inhibition, any mismatch between the RBD used in the assay and the epitopes to which the RBD-specific antibodies in the sample are directed can diminish their potential as a substitute assay for live-virus NTs. Similarly to the loss in sensitivity when detecting mAbs after Sotrovimab administration, sVNTs were significantly reduced in their ability to identify the antibody response in individuals after primary Omicron infection, whose antibodies are exclusively directed against the antigenically changed RBD of the Omicron variant [21]. Furthermore, conformational differences in the RBD used as the respective assays´ target antigen could contribute to the inter-assay variations we observed in the present study. 

Since Anti-Spike immunoassays (QuantiVAC, TrimericS IgG) use the entire Spike protein as the target antigen, this may explain their improved ability to assess Sotrovimab levels, as indicated by the stronger correlation between the measured BAU/mL levels and the NT titers (Figure 4). However, the evaluated commercial immunoassays still use the ancestral Spike (or RBD) of the ancestral wild-type as the antigen. In principle, such a mismatch could have affected the correlation between the antibody levels quantified by these assays and Omicron-specific NT titers. Thus, further studies should evaluate whether novel immunoassays containing the Omicron variant´s Spike protein display a better correlation. Nonetheless, the significant inter-assay variations we observed exclusively occurred in immunoassays that were not yet adapted to the Omicron variant.

A limitation of this study is that we did not determine the clinical effectiveness of Sotrovimab as PrEP, but rather focused on comparing the measured antibody levels and live-virus NT results. The effectiveness of Sotrovimab against the currently circulating variants has decreased [8,9,10,22]. However, SARS-CoV-2 is currently mutating, so the return of virus variants that are more susceptible to Sotrovimab cannot be excluded, as recent studies suggest for variants BA2.75.2, BQ1.1 and XBB1.5 [22,23,24,25]. Cut-off levels for future variants with increased or similar susceptibility to Sotrovimab can be easily assessed using adapted NTs. In addition, a higher dosing of Sotrovimab is currently being tested in clinical studies and may provide a feasible strategy to overcome their reduced effectiveness, further emphasizing the need for a monitoring strategy, as outlined in this study [26].

In conclusion, this study shows significant variations among commercial antibody assays in measuring Sotrovimab concentration in human serum samples after infusion. Nonetheless, although the correlation of antibody levels measured by immunoassays and live-virus NT titers was limited for certain tests, immunoassays present a viable alternative for guiding Sotrovimab redosing in PrEP, as long as conservative cut-offs are employed. However, as in other fields of laboratory diagnostics, specific SARS-CoV-2 immunoassays have to be carefully selected for the respective application area. In this regard, our study demonstrates that, compared to Spike-specific IgG antibody assays, sVNTs are limited in their ability to guide Sotrovimab redosing in PrEP.

## Figures and Tables

**Figure 1 viruses-15-01624-f001:**
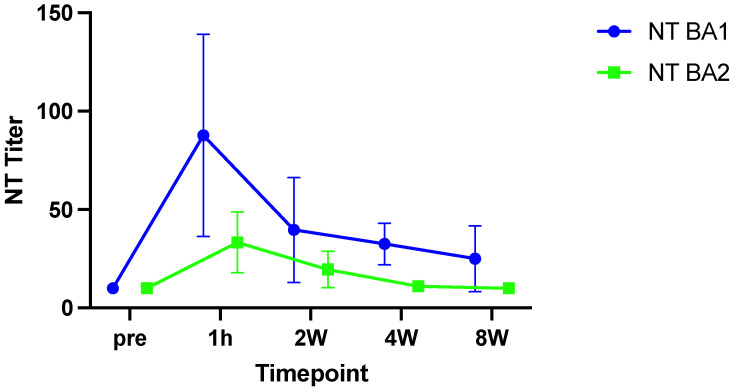
Live virus neutralization tests for BA1 (blue) and BA2 (green). Reduced neutralization capacity is observed for Sotrovimab against the BA2 subvariant. Mean and standard deviation are shown. NT: neutralization test.

**Figure 2 viruses-15-01624-f002:**
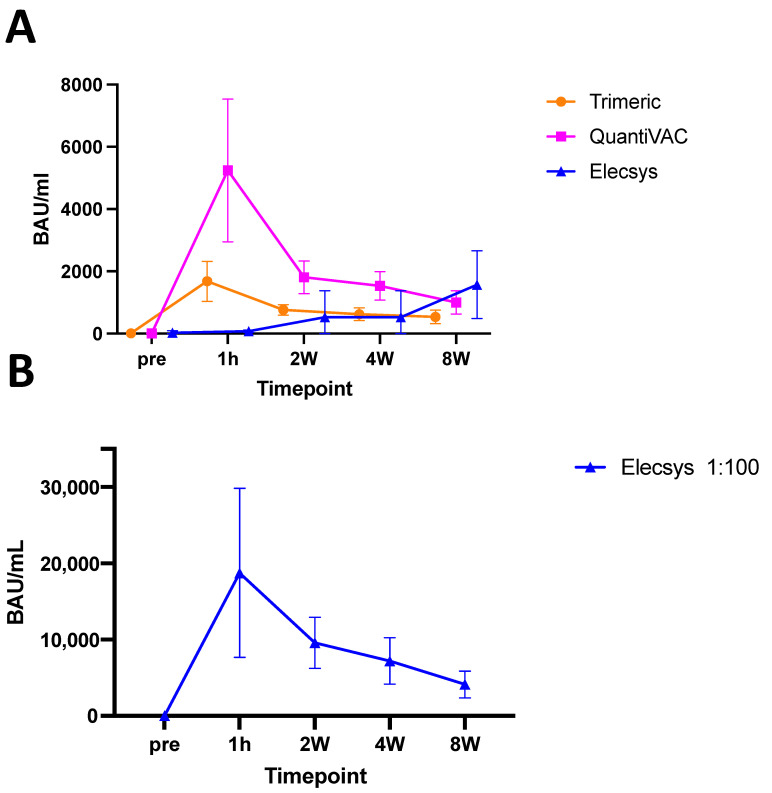
Antibody measurements by (**A**) TrimericS CLIA (Orange), QuantiVAC ELISA (pink), Elecsys systems (blue) and (**B**) Elecsys after 1:100 dilution. Initial increase after infusion is followed by reduced antibodies over time, as expected. For the Elecsys system, a prior 1:100 dilution is necessary due to a presumed prozone effect. Shown are mean and standard deviation.

**Figure 3 viruses-15-01624-f003:**
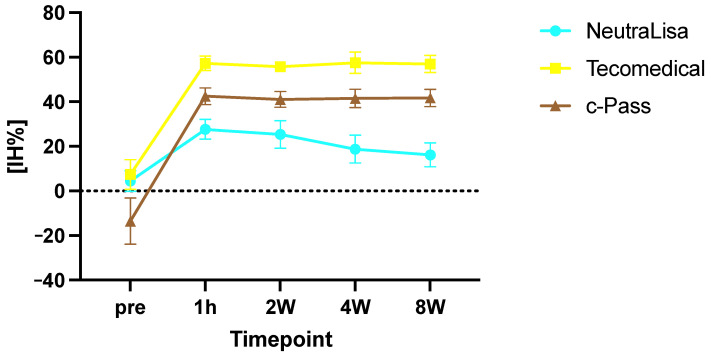
Surrogat neutralization test results determined with the NeutraLISA (blue), Teicomedical (yellow) and cPass system (brown). Signal inhibition (IH%) increased after infusion and remained high over the entire observational time. Shown are mean and standard deviation.

**Figure 4 viruses-15-01624-f004:**
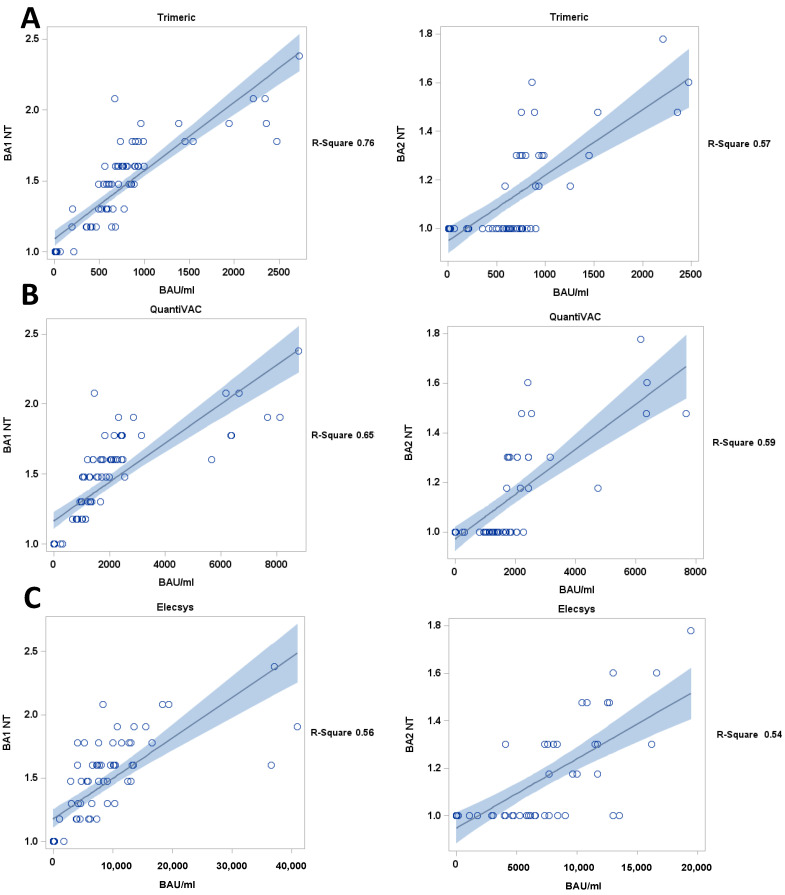
Correlation of antibody levels (BAU/mL) and live-virus BA1 and BA2 NT titers, (**A**) TrimericS and (**B**) QuantiVAC an (**C**) Elecsys (diluted samples) assay. black line: regression line, blue: confidence limits; Y axis = BA1 NT titers, BA2 NT titers.

**Figure 5 viruses-15-01624-f005:**
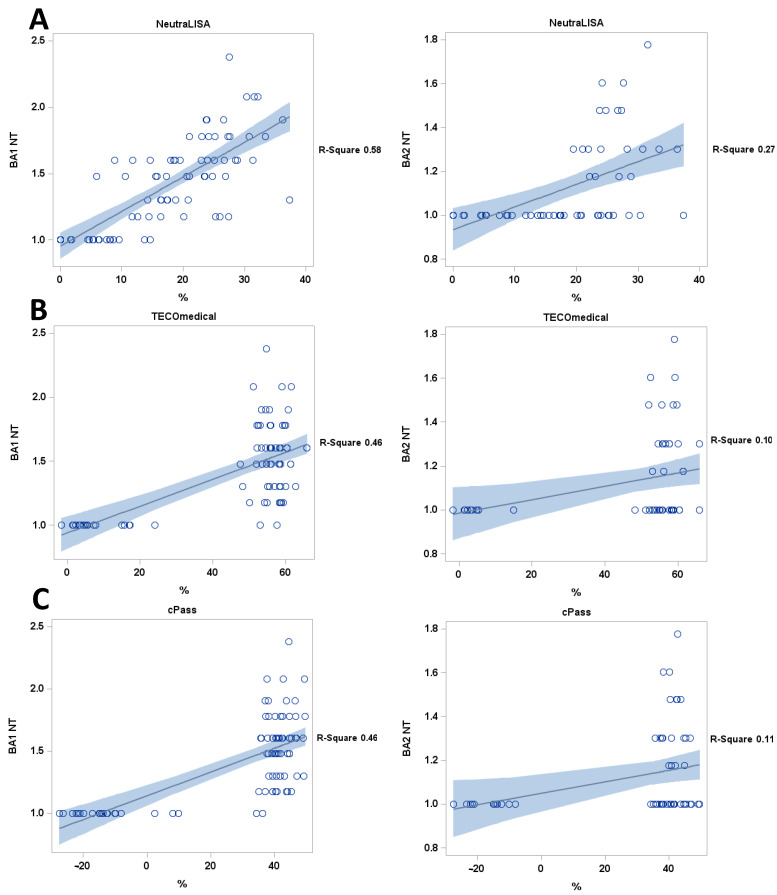
Correlation of sVNTs (**A**) NeutraLISA, (**B**) TECOmedical and (**C**) cPass with live-virus NT titers. All sVNTs show poor correlation with live-virus NT results. black line: regression line, blue: confidence limits; blue dotted blue lines: prediction limits; Y axis = BA1 NT titers, BA2 NT titers.

**Figure 6 viruses-15-01624-f006:**
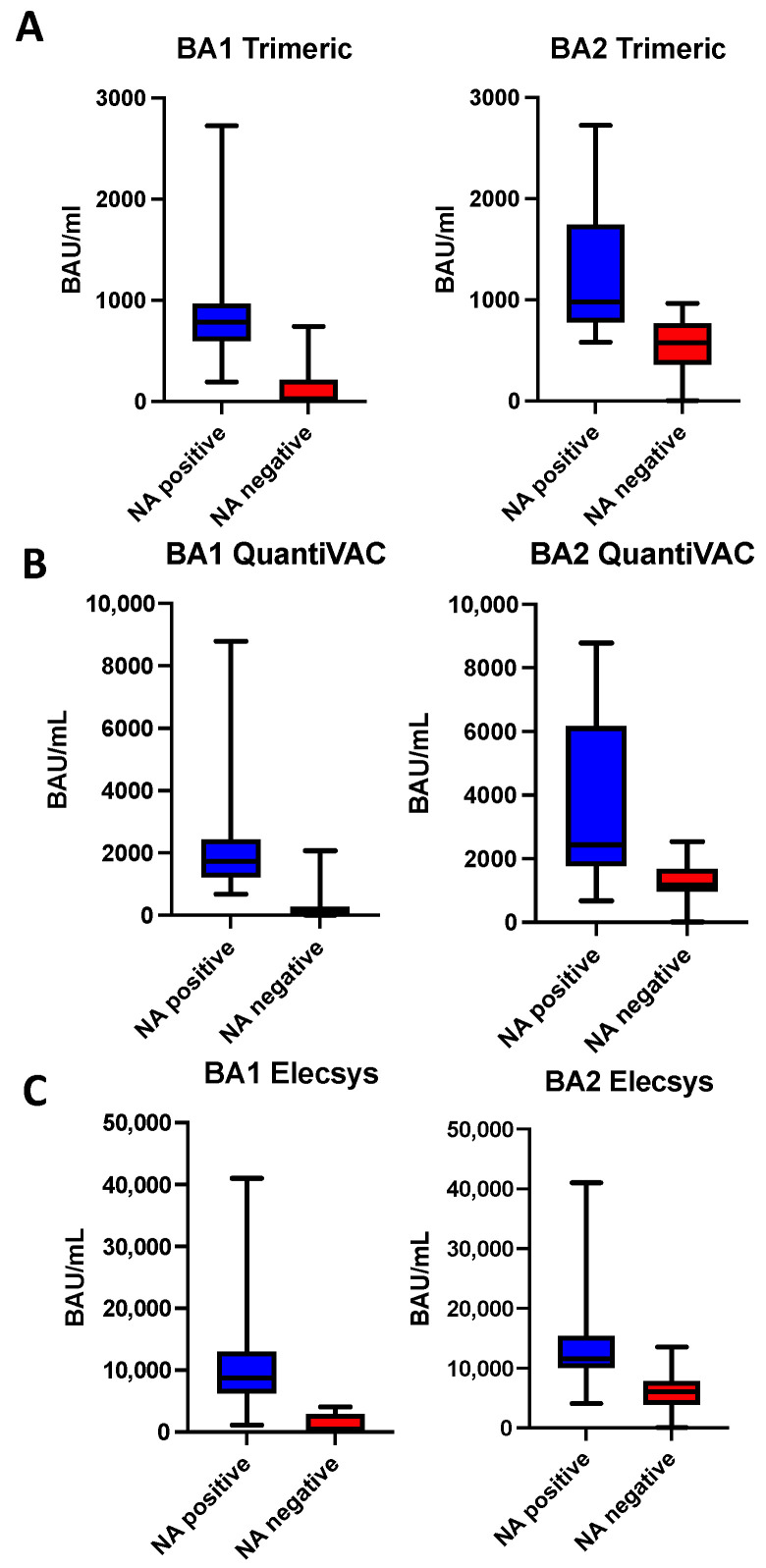
Measured antibody levels (BAU/mL) with the Elecsys (**A**), TrimericS (**B**) and QuantiVAC (**C**) assays stratified by live-virus NT results. Shown are median, first and third quartile.

**Table 1 viruses-15-01624-t001:** Patient characteristics. CNI: calcineurin inhibitor; MPA: mycophenolic acid; Tx: transplant.

Patient Characteristics	
Age (years)	60.5 (41–76)
Sex (female)	12 (60%)
Number of previous vaccine doses (n)	3 (3–4)
Immunosuppression	
CNI + MPA +S teroids	16 (80%)
Belatacept + MPA + Steroids	4 (20%)
TX vintage (years)	3.5 (1–22)
Number of previous transplantations (n)	1 (1–3)

## Data Availability

The datasets used and analyzed during the current study are available from the corresponding author on reasonable request.

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
