# Peer review of "Monitoring of Sotrovimab-Levels as Pre-Exposure Prophylaxis in Kidney Transplant Recipients Not Responding to SARS-CoV-2 Vaccines"

_viruses, 2023, doi:10.3390/v15081624_

Round 1
Reviewer 1 Report
The authors have conducted a study in which 20 kidney transplant recipients were administered 500mg of Sotrovimab, and their antibody levels were monitored over eight weeks using different detection methods. The results indicate that Sotrovimab has a neutralizing effect against the Omicron variant of SARS-CoV-2. However, there is variation in the antibody levels detected by different methods, and the correlation between antibody levels and neutralizing capability also varies. The authors concluded that current antibody measurement platforms may not be suitable for monitoring the neutralizing effect of Sotrovimab against SARS-CoV-2.
To further improve this study, I recommend the following:
1. Figure 2 shows that different reagents from different manufacturers might produce different Binding Antibody Units (BAU) values for the same serum. This point is mentioned in the discussion, but it warrants further exploration. Specifically, is there any variation between batches for the same specimen?
2. The presentation of data in Figures 4 and 5 could be improved to make the data more distinct.
3. Figure 5 indicates a weaker correlation between various surrogate neutralization tests (NeutraLISA, TECOmedical, and cPass) and live virus neutralization titers. This suggests a substantial discrepancy between the results of these surrogate tests and the live virus neutralization titers. As a result, relying solely on these surrogate tests to evaluate the neutralizing ability of antibodies might not accurately predict the protective capacity of antibodies against live virus infection.
4. The reagent development, which did not specifically target the BA variant, could have resulted in numerical bias. This point should be elaborated upon and discussed further in the discussion section.
I suggest having the manuscript reviewed by a professional language editing service or a native English speaker to ensure the language quality is up to the standard of a scientific publication.
Author Response
- Figure 2 shows that different reagents from different manufacturers might produce different Binding Antibody Units (BAU) values for the same serum. This point is mentioned in the discussion, but it warrants further exploration. Specifically, is there any variation between batches for the same specimen?
-We agree with the Reviewer that there were significant manufacturer-dependent differences in the quantification of Spike-specific IgG antibodies after the infusion with sotrovimab. Interestingly, this variation showed a different pattern for the TrimericS and the QuantiVac assay than in samples after SARS-CoV-2 infection and vaccinations, where multiple studies have previously described higher BAU/mL levels in the TrimericS assay. Differences in the measured BAU/mL levels between those assays after sotrovimab infusion are most likely related to the conformational differences of the Spike protein used in the assays as target antigens, which may significantly affect the specific binding of the monoclonal antibodies contained in sotrovimab.
We now included this information in the revised version of the manuscript (Lines 270-276).
By performing additional dilution steps, we excluded that a prozone effect also affected the TrimericS assay and lower BAU/ml levels assessed by this assay (data not shown). Since the samples were tested with the same intra-assay batches after multiple dilution steps, we consider it unlikely that differences in the batches caused the variation we observed in the study.
- The presentation of data in Figures 4 and 5 could be improved to make the data more distinct.
- The figures four and five have been adjusted to improve the visualization of the data.
- Figure 5 indicates a weaker correlation between various surrogate neutralization tests (NeutraLISA, TECOmedical, and cPass) and live virus neutralization titers. This suggests a substantial discrepancy between the results of these surrogate tests and the live virus neutralization titers. As a result, relying solely on these surrogate tests to evaluate the neutralizing ability of antibodies might not accurately predict the protective capacity of antibodies against live virus infection.
-Indeed, we agree with the Reviewer that it was an interesting finding of this study that certain sVNTs were significantly reduced in their ability to assess neutralizing antibodies and are therefore not well suited as substitute assays for live-virus NTs after sotrovimab infusion (in contrast to individuals after wild virus infections and vaccinations). To elaborate on this point, we included additional text segments in the material and methods (Lines 96 to 99) and the discussion (297 to 305), as recommended (also see Comment 2 from Reviewer 2).
- The reagent development, which did not specifically target the BA variant, could have resulted in numerical bias. This point should be elaborated upon and discussed further in the discussion section.
-We agree with the Reviewer that most commercial immunoassays have not yet been adapted to the Omicron variant and still contain the Spike (or the Receptor-Binding-Domain) proteins of the ancestral wild-type. In principle, such a mismatch could affect the correlation between Spike-specific antibodies quantified with these assays and NT titers assessed by Omicron-specific live-virus NTs. Therefore, we are currently performing a study to analyze whether the correlation with live-virus NT titers is improved for commercial immunoassays that use the Omicron variant's Spike protein rather than the wild-type one.
However, as a main result of our study we demonstrate a significant variation among different immunoassays to measure mAbs after sotrovimab infusion, which have all not yet been adapted to the Omicron variant.
As the Reviewer recommended, we now included this important point in the discussion (LINES 309-316).
Reviewer 2 Report
The authors test several assays for measuring Sotrovimab concentrations in the blood of treated patients. They find that the surrogate virus assays are not accurate. QuantiVAC and TrimericS immunoassays were acceptable while 3 other commercial assays were not.
The measurement of Sotrovimab levels in treated patients seems like a useful thing to do assuming that the antibody remains in use. It’s hard to see how it could be of much use as its inactive against current variants XBB and others.
The experiments seem to be carefully done and the results are reliable. There are problems with the text that need to be addressed as detailed below. There are mistakes in spelling and small mistakes that need to be fixed. Overall, the paper is aggravating to read because of several unclear statements and points that are not clearly explained.
Specific points
1. Line 79. This is headed “live virus tests” in the plural yet there is only one such assay used. This should be singular so that readers and reviewers do not waste time searching for other live virus assays.
2. It is difficult to understand the different assays that are being tested. They are classified as SNVs (cPASS, TECO and neutraLISA) and 2 immunoassays. The authors’ major conclusion is that SNVs are no good for Sotrovimab. They seem to say that the immunoassays are acceptable but the statement in the final paragraph about this is unintelligible. Just state whether immunoassays are useful or not. It is not helpful just conclude that SNVs are no good as the only take-home message of the paper.
3. The authors offer no explanation that would guide a nonspecialist to understand the principle of the assays used. They seem to think readers should know this information. They need to state what these different assays are and how they work.
4. The authors offer no explanation as to why the SNVs don’t work. Most likely, it’s because Sotrovimab doesn’t bind to the spend in the RBD on which these assays are based. This would explain it and why the immunoassay which uses the whole spike does work.
5. There does not seem to be any reason to use patient sera in this study. Why not just use the different assays with pure Sotrovimab? Wouldn’t that show which assays work and which don’t? It would allow absolute measurements to be made regarding the amount of protein in the assays to state the concentration of antibody that can be detected in each assay. The title about kidney transplant patients seems to be irrelevant.
6. Figures 4 and 5. The Y axis is unintelligible due to some sort of typographical problem.
7. Line 79. "Surrogate" is misspelled.
8. Line 65. It says there are 3 SVAs and then lists just two.
9. Line 57. Use of the words “conveyed” in regard to administration of antibody makes no sense.
10. Line 71. The abbreviation “BAU” needs to be defined here, not later in the text as is currently the case.
English is good overall but a few mistakes need to be corrected.
Author Response
- Line 79. This is headed “live virus tests” in the plural yet there is only one such assay used. This should be singular so that readers and reviewers do not waste time searching for other live virus assays.
-The headline was corrected.
- It is difficult to understand the different assays that are being tested. They are classified as SNVs (cPASS, TECO and neutraLISA) and 2 immunoassays. The authors’ major conclusion is that SNVs are no good for Sotrovimab. They seem to say that the immunoassays are acceptable but the statement in the final paragraph about this is unintelligible. Just state whether immunoassays are useful or not. It is not helpful just conclude that SNVs are no good as the only take-home message of the paper.
-We agree with the Reviewer that the final paragraph required revision. We now clarified our final statement (LINES 329-335).
- The authors offer no explanation that would guide a nonspecialist to understand the principle of the assays used. They seem to think readers should know this information. They need to state what these different assays are and how they work.
-As recommended by the Reviewer, we included additional information on the test principle of the different immunoassays (LINES 109-113).
- The authors offer no explanation as to why the SNVs don’t work. Most likely, it’s because Sotrovimab doesn’t bind to the spend in the RBD on which these assays are based. This would explain it and why the immunoassay which uses the whole spike does work.
-As the Reviewer recommended, we included more information on the sVNTs test principle and thoroughfully discussed possible functional explantions why the sVNTs were reduced to detect neutralizing antibody responses after Sotrovimab infusion.
- There does not seem to be any reason to use patient sera in this study. Why not just use the different assays with pure Sotrovimab? Wouldn’t that show which assays work and which don’t? It would allow absolute measurements to be made regarding the amount of protein in the assays to state the concentration of antibody that can be detected in each assay. The title about kidney transplant patients seems to be irrelevant.
-This is correct, to test the useability of the different test assays solely pure sotrovimab would have been sufficient. In this study we were aiming for a monitoring strategy of sotrovimab in immunocompromised patients in an in-vivo setting and describe its dynamics in a PreP use.
- Figures 4 and 5. The Y axis is unintelligible due to some sort of typographical problem.
- The figures have been updated.
- Line 79. "Surrogate" is misspelled.
-We were not able to find the adressed misspelling. Line 79 is a headline as mentioned in question one by the Reviewer (“live virus test”).
- Line 65. It says there are 3 SVAs and then lists just two.
- This was corrected and all three are mentioned
- Line 57. Use of the words “conveyed” in regard to administration of antibody makes no sense.
-The word “conveyed” was replaced.
- Line 71. The abbreviation “BAU” needs to be defined here, not later in the text as is currently the case.
- The abbreviation was added.
Round 2
Reviewer 1 Report
The author responded diligently to these suggestions, making necessary modifications and additions to the manuscript. Their revisions have significantly enhanced the quality and clarity of the paper, aligning it with the requirements of our publication.